



# Observations of ozone-poor air in the Tropical Tropopause Layer

Richard Newton[1], Geraint Vaughan[1], Eric Hintsa[2], Michal T. Filus[3], Laura L. Pan[4], Shawn Honomichl[4], Elliot Atlas[5], Stephen J. Andrews[6], and Lucy J. Carpenter[6]

[1]National Centre for Atmospheric Science, The University of Manchester, UK
[2]NOAA ESRL Global Monitoring Division, Boulder, CO, USA
[3]Centre for Atmospheric Science, Department of Chemistry, University of Cambridge
[4]National Center for Atmospheric Research, Boulder, CO, USA
[5]Department of Atmospheric Sciences, RSMAS, University of Miami, Miami, FL, USA
[6]Department of Chemistry, Wolfson Atmospheric Chemistry Laboratories, University of York

*Correspondence to:* G. Vaughan
geraint.vaughan@manchester.ac.uk

**Abstract.** Ozonesondes reaching the tropical tropopause layer (TTL) over the West Pacific have occasionally measured layers of very low ozone concentrations—less than 15 ppbv—raising the question of how prevalent such layers are and how they are formed. In this paper we examine aircraft measurements from the ATTREX, CAST and CONTRAST campaigns based in Guam in January–March 2014 for evidence of very low ozone concentrations and their relation to deep convection. The study

builds on results from the ozonesonde campaign conducted from Manus Island, Papua New Guinea, as part of CAST, where ozone concentrations as low as 12 ppbv were observed between 100 and 150 hPa downwind of a deep convective complex.

TTL measurements from the Global Hawk unmanned aircraft show a marked contrast between the hemispheres, with mean ozone concentrations in profiles in the Southern Hemisphere between 100 hPa and 150 hPa of between 10.5 ppbv and 14.2 ppbv. By contrast, the mean ozone concentrations in profiles in the Northern Hemisphere were always above 15 ppbv and normally

above 20 ppbv at these altitudes. The CAST and CONTRAST aircraft sampled the atmosphere between the surface and 120 hPa, finding very low ozone concentrations only between the surface and 700 hPa; mixing ratios as low as 7 ppbv were regularly measured in the boundary layer, whereas in the free troposphere above 200 hPa concentrations were generally well in excess of 15 ppbv. These results are consistent with uplift of almost-unmixed boundary layer air to the TTL in deep convection. An interhemispheric difference was found in the TTL ozone concentrations, with values <15 ppbv measured extensively in the

Southern Hemisphere but seldom in the Northern Hemisphere. This is consistent with a similar contrast in the low-level ozone between the two hemispheres found by previous measurement campaigns. Further evidence of a boundary layer origin for the uplifted air is provided by the anti-correlation between ozone and halogenated hydrocarbons of marine origin observed by the three aircraft.

## 1 Introduction

Air entering the stratosphere in the Brewer-Dobson circulation originates in the Tropical Tropopause layer (TTL), a region between around 13 and 17 km altitude with characteristics intermediate between the highly convective troposphere below and



the stratified stratosphere above (Holton et al., 1995; Highwood & Hoskins, 1998; Folkins et al., 1999; Gettelman & Forster, 2002; Fueglistaler et al., 2009). The TTL lies above the main convective outflow (10–13 km) and although deep convection can reach, and even overshoot the tropopause (e.g. Frey et al., 2015), the region is not well-mixed and both radiative and large-scale dynamical processes influence its structure and composition (Fueglistaler et al., 2009). A key question about the TTL is whether

deep convection is nevertheless capable of lifting very short-lived halogenated species near enough to the tropopause that their breakdown products reach the stratosphere and contribute to ozone destruction. In this paper we use ozone measurements from a unique aircraft campaign to investigate the uplift of air from near the Earth's surface to the TTL.

The oceanic Tropical Warm Pool in the Western Pacific and Maritime Continent is marked by very warm surface temperatures (>27°C) and is therefore able to sustain widespread deep convection. Above the Warm Pool a number of ozonesonde ob-

servations have shown very low ozone concentrations near the tropopause (Kley et al., 1996; Heyes et al., 2009; Rex et al., 2014; Newton et al., 2016), possibly indicative of uplift of near-surface air by deep convection. Unfortunately, accurate ozonesonde measurements in this part of the atmosphere are very difficult as the sondes produce a poorly-characterized background current which can be half the measured signal in the TTL (Vömel & Diaz, 2010; Newton et al., 2016). Nevertheless, even after taking this into account, there is evidence of ozone mixing ratios < 15 ppbv occurring just below the tropopause. These are found in

localized regions, or bubbles, generally associated with deep convection.

The first evidence of low-ozone bubbles in the TTL was provided by the CEPEX campaign (Kley et al., 1996), where near-zero ozone concentrations were reported between the Solomon Islands and Christmas Island. These ozonesondes were affected by the background current problem, and after Vömel & Diaz (2010) reanalyzed the data with a more representative background current correction, the minimum measured ozone concentration was ~8 ppbv. Ozone concentrations <15 ppbv were found by

Heyes et al. (2009) in Darwin, Australia in the ACTIVE campaign in 2005–6, and on the TransBrom cruise of 2009 in the West Pacific (Rex et al., 2014). More recently, Newton et al. (2016) presented TTL ozone measurements as low as 12 ppbv from Manus Island, Papua New Guinea (2.07°S, 147.4°E); we discuss these measurements in more detail in section 3.

The mechanism for producing low-ozone bubbles in the TTL are not fully understood. Clearly, uplift in deep convection is the underlying cause, but deep convection is a turbulent process and air entering at the surface would be expected to mix with

its surroundings during ascent. Noting that the minimum ozone concentrations observed in the TTL above Darwin were too low to originate in the boundary layer locally, Heyes et al. (2009) proposed long-range transport in the TTL from a 'hot-spot' region north-east of New Guinea. Newton et al. (2016) found that the minimum concentrations measured over Manus were only consistent with ozone measurements in the lowest 300 m over the island, suggesting uplift of air from near the surface to the TTL with little or no mixing (see below). Clearly, there is a need to corroborate these sporadic ozonesonde observations

with other measurements and to determine how widespread these bubbles of low-ozone air are over the Warm Pool. This is the purpose of the present paper.

During January-March 2014, a coordinated aircraft campaign was conducted from Guam (13.44°N, 144.80°E) to measure the atmosphere over the Tropical Warm Pool in unprecedented detail. Three aircraft were involved:

- the NASA Global Hawk unmanned aircraft, as part of the Airborne Tropical Tropopause Experiment, ATTREX (Jensen
et al., 2017)



- the NCAR Gulfstream V research aircraft, as part of the Convective Transport of Active Species in the Tropics Experiment, CONTRAST (Pan et al., 2017)

- the UK Facility for Airborne Atmospheric Measurement (FAAM) BAe 146 aircraft, as part of the Coordinated Airborne Studies in the Tropics (CAST) experiment, (Harris et al., 2017)

Together these three aircraft were able to sample the tropical atmosphere from the surface to the lower stratosphere, enabling detailed measurements of the inflow and outflow of deep convection and the environment in which it formed. Of particular interest to this paper is the Global Hawk, which extensively sampled the TTL. With a typical flight range of 16,000 km and duration of up to 24 hr, the aircraft continuously executed profiles between 45,000 ft (13.7 km) and 53,000–60,000 ft (16.2–18.3 km) (Jensen et al., 2017). Thus it was able to gather a wealth of profiles of ozone and other gases through the TTL both in the Northern and Southern Hemisphere. The NCAR Gulfstream V aircraft sampled mainly in the Northern Hemisphere, between sea level and 15 km altitude (Pan et al., 2017) although some measurements were also made in the Southern Hemisphere, most notably on a flight to 20°S on 22 February 2014. The FAAM aircraft sampled the lower atmosphere, from the ground to 10 km altitude but with most of the measurements in the boundary layer. These measurements were almost all in the Northern Hemisphere.

After describing key aircraft instrumentation, this paper first presents the salient results from the CAST ozonesonde campaign, which were described in detail by Newton et al. (2016). We then introduce the Global Hawk ozone profiles, concentrating on one flight that sampled well into the Southern Hemisphere from Guam. After a brief consideration of the other Global Hawk flights, we examine some of the other chemical species measured by the three aircraft before ending with discussions and conclusions.

## 2   Instrumentation

Ozone was measured in situ by all three aircraft in the CAST, CONTRAST and ATTREX campaigns. The FAAM BAe 146 carried a Thermo Fischer Model 49C UV absorption photometer, which had an uncertainty of 2% and a precision of 1 ppbv for 4 s measurements (Harris et al., 2017). Ozone on the Gulfstream V aircraft was measured using the NCAR Chemiluminescence instrument, which uses the chemiluminescent reaction between nitric oxide and ozone. The detection limit was below 0.1 ppbv, and its accuracy within 5% for the entire range of ozone measurements made during CONTRAST (Ridley et al., 1992; Pan et al., 2015, 2017).

On the Global Hawk, the UCATS instrument (UAS (Unmanned Aerial System) Chromatograph for Atmospheric Trace Species) provided measurements of ozone, plus nitrous oxide ($N_2O$), sulphur hexafluoride ($SF_6$), hydrogen ($H_2$), carbon monoxide (CO) and methane ($CH_4$) (Jensen et al., 2017). Ozone in the UCATS unit is measured by two Model 205 UV photometers from 2B Technologies (Boulder, Colorado) modified for high altitude operation. The first was mounted inside the UCATS package, whilst a second, newer Model 205 photometer was added to the front panel of the UCATS. Both instruments were modified to include stronger pumps (KNF model UNMP-830), scrubbers with magnesium oxide (MgO) coated screens,





and pressure sensors with a range from 0 hPa to >1000 hPa (Honeywell ADSX series). The model 205 is a 2-channel photometer, with the flow continuously split between the unscrubbed (ambient) air into one cell and scrubbed (ozone-free) air into the other for measurement by the Beer-Lambert law absorption of 253.7 nm radiation from a Hg lamp. Flow is switched every two seconds and data recorded at this rate for the newer instrument but averaged to 10 s in the older model. The instruments are

calibrated on the ground against a NIST-certified calibration system (Thermo Electron, Inc.) before and after every flight. In all cases, the slope of the regression line between the instrument and calibrator data was within 1% of unity and the offset less than 2 ppbv (usually <1 ppbv) at ambient pressure and room temperature. However, in-flight comparisons on earlier ATTREX missions between the 2B instruments and the NOAA ozone photometer (Gao et al., 2012) revealed a possible negative bias of up to 5 ppbv at low ozone concentrations.

In addition to ozone data, selected whole air sampler (WAS) data are used to identify convective influence. Whole air samplers were on board all three aircraft, measuring a large array of compounds. All three aircraft measured dimethyl sulphide ($(CH_3)_2S$), iodomethane ($CH_3I$), dichloromethane ($CH_2Cl_2$), bromochloromethane ($CH_2BrCl$), trichloromethane ($CHCl_3$), dibromochloromethane ($CHBr_2Cl$) and tribromomethane ($CHBr_3$). The WAS samples collected on board the BAe 146 were analyzed typically within 72 hours of collection, with gas chromatography-mass spectrometry (GC-MS; Agilent 7890 GC,

5977 Xtr MSD) (Andrews et al., 2016; Harris et al., 2017). The CONTRAST and ATTREX whole air samplers were analyzed for a much larger array of compounds, and were also analyzed by the GC-MS. The samples were split between an Agilent HP-AL/S PLOT (porous layer open tubular) column with a flame ionization detector, and the remaining sample was split again between an electron capture detector and an Agilent 5975 GC-MSD (gas chromatograph—mass selective detector) (Schauffler et al., 1999; Apel et al., 2003; Andrews et al., 2016).

## 3 Manus ozonesondes

Thirty-three ozonesondes were launched from Manus Island during February 2014 as part of CAST. A salient result of the campaign was further insight into the background current: where this quantity was $\lesssim$50 nA, a constant offset was subtracted from the measured current, but when the background current was larger, a hybrid correction was applied which decreased with height (Newton et al., 2016). These procedures gave good agreement with nearby ozone measurements on the Gulfstream V on

5 and 22 February, verifying the use of ozonesondes to measure very low ozone concentrations near the tropical tropopause.

Another salient result from the ozonesonde measurements was the low-ozone event in the TTL between 18 and 23 February, visible in figure 1, where measured ozone was as low as 12 ppbv. As Folkins et al. (2002) argued, the only region of the tropical troposphere able to generate ozone concentrations ≤20 ppbv is near the surface, so this air mass is likely to be of recent boundary layer origin. Ozone concentrations through most of the boundary layer over Manus in this period were higher than in the TTL; only in the bottom 300 m of the profiles did the ozonesondes measure < 20 ppbv, with concentrations at

the ground around 8–15 ppbv (Newton et al., 2016). This suggests either that the ozone-poor air was lifted from near the surface, or that boundary layer ozone concentrations in the uplift region were lower than over Manus. A back-trajectory analysis of the low-ozone bubble with the on-line HYSPLIT model (Stein et al., 2015) indicated that the origin of the low





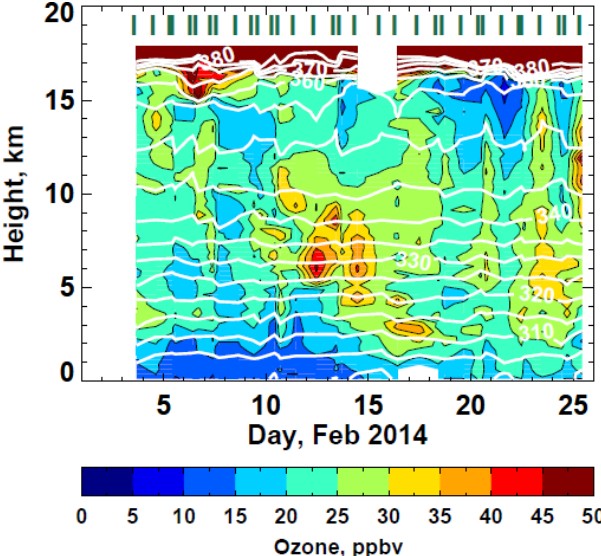

**Figure 1.** Contour plot of ozone concentrations from the Manus ozonesondes. The white lines are potential temperature isolines. Note the very low concentrations in the upper troposphere around 21–22 February. Green bars along the top denote launch times of ozonesondes. For further details see Newton et al. (2016).

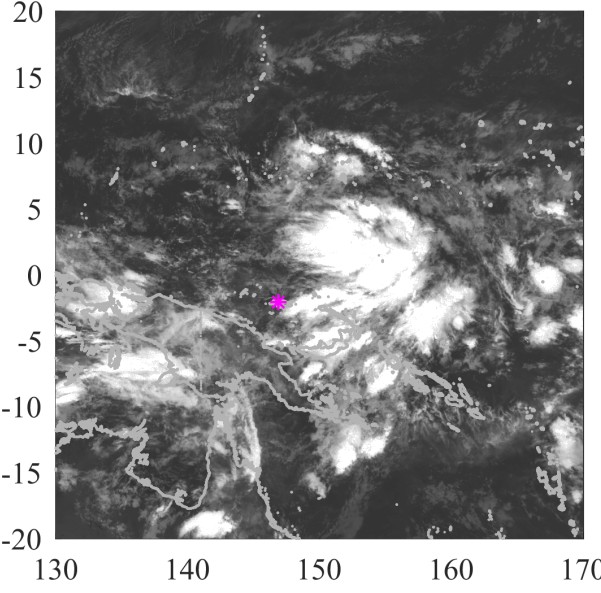

**Figure 2.** MTSAT infrared satellite image from 19 February 2014 at 18:00 UTC, showing the convection to the east of Manus Island (pink star) that was determined to be the origin of the low-ozone air in the TTL above Manus.



concentrations of ozone was a mesoscale convective system to the east of Manus Island that uplifted air from the lower troposphere into the tropopause layer (see figure 2), combined with a strong easterly jet that advected the air towards Manus Island. Unfortunately, ozone measurements were not available in this area at this time so the altitude from which the ozone-poor air was lifted remains an open question. To examine whether further examples of ozone-poor layers were encountered during

the CAST/CONTRAST/ATTREX campaign, we now examine the Global Hawk observations in the TTL during February and March 2014.

## 4  Global Hawk measurements

### 4.1  ATTREX flights

The Global Hawk measured in the same altitude range as the layer of ozone-poor air above Manus Island: the aircraft performed

ascents and descents between 150 hPa (13.6 km) and 100 hPa–75 hPa (16.1 km–18.0 km), depending on fuel load. The ascent rate was slow, of the order of 45 minutes to complete at an average vertical velocity of ~0.5 m·s$^{-1}$, but the descent rate was much quicker, of the order of 5–10 minutes to complete at ~4 m·s$^{-1}$. Only the ascent data are used in this study as the descent was found to be too quick for reliable ozone measurements.

In total, six research flights were flown by the Global Hawk from Guam during the ATTREX campaign. The first two, RF01

and RF02, were on 12 and 16 February when the CAST and CONTRAST campaigns were active, but there was a gap of fifteen days between the second and third flights as the aircraft developed a problem; the final four flights, RF03, RF04, RF05 and RF06 were on 4, 6, 9 and 11 March respectively—after CAST and CONTRAST had finished. The two transfer flights to and from Anderson Flight Research Center on 16 January and 13 March made few measurements in the West Pacific region and are not considered here.

Flight RF01 on 12–13 February focused on the composition, humidity, clouds and thermal structure of the Northern Hemisphere part of the Warm Pool region. Convection was situated mostly around the Maritime Continent on this day (figure 3), with no notable convection around Guam. The second flight, RF02 occurred on 16–17 February with similar scientific objectives to RF01. As a result of a satellite communications problem, the aircraft was required to stay in line-of-sight contact with the airbase in Guam, and consequently the aircraft flew in a small area of airspace close to the island. On this day, convection

was visible to the southeast of Guam in the MTSAT satellite imagery (figure 4).

The third flight took place after a two-week hiatus on 4–5 March. Its objectives were to sample the outflow of tropical cyclone Faxai, which developed in the region in the previous few days, with vertical profiles performed to observe the outflow cirrus cloud from the cyclone. Apart from tropical cyclone Faxai, the majority of the convection was in the Southern Hemisphere around Papua New Guinea (figure 5).

Flight RF04 took place on 6–7 March. Tropical cyclone Faxai had dissipated by this time, leaving a dearth of convection in the Northern Hemisphere; the most convectively active region was around Papua New Guinea (figure 6). RF05 surveyed the Southern Hemisphere on 9–10 March, measuring the lowest ozone concentrations observed by the Global Hawk during the ATTREX campaign. This flight is discussed in detail in the next section. The final research flight, RF06, took place on



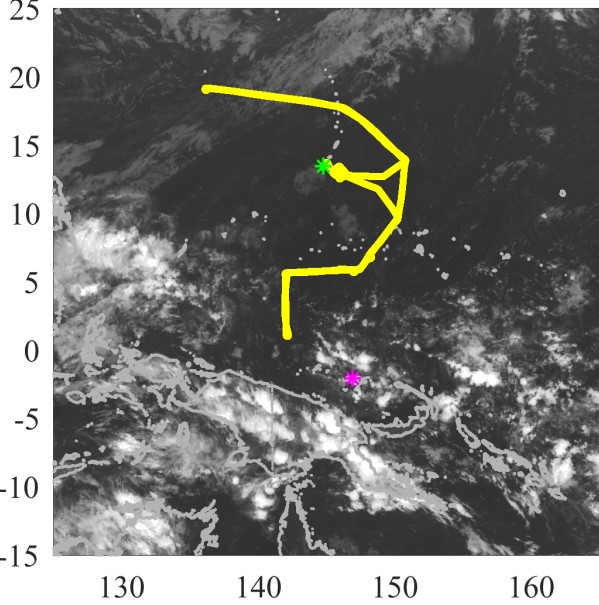

**Figure 3.** MTSAT infrared satellite image from 12 February at 12:00 UTC, coincident with flight RF01 (yellow track). Green asterisk denotes location of Guam; magenta asterisk that of Manus Island. Convection is centred mostly around the Maritime Continent on this day.

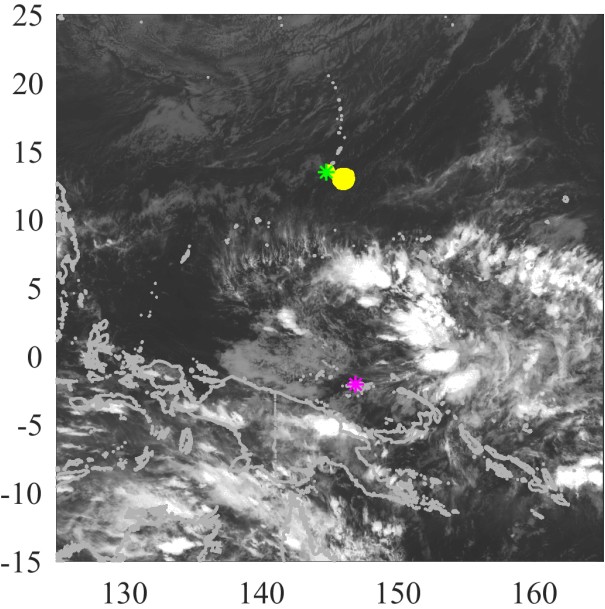

**Figure 4.** As figure 3 but for 16 February at 12:00 UTC coincident with flight RF02. A band of convective activity is visible to the southeast of Guam.





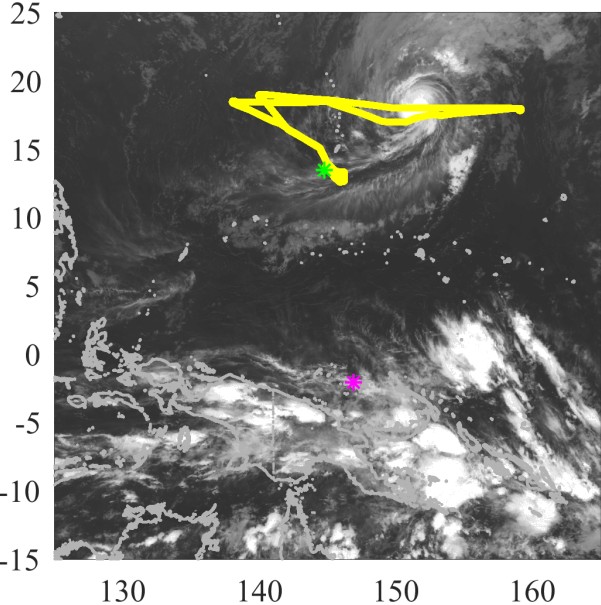

**Figure 5.** Satellite image of March 4 at 12:00 UTC, coincident with flight RF03. Cyclone Faxai is visible to the northeast of Guam.

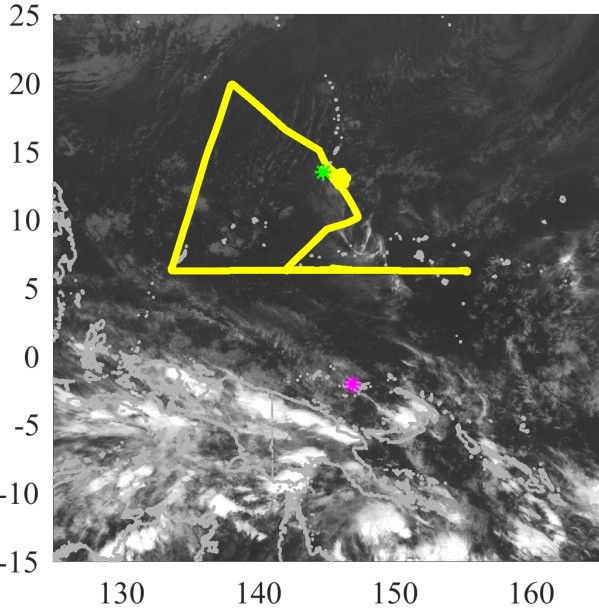

**Figure 6.** Satellite image of March 6 at 12:00 UTC, coincident with flight RF04. Convection is minimal in the Northern Hemisphere, and is concentrated mostly in the Southern Hemisphere.

11–12 March, surveying latitudes north of 10°N either side of the subtropical jet, and is outside the scope of this paper. A full description of the ATTREX flights and meteorological conditions encountered can be found in Jensen et al. (2017).



## 4.2 ATTREX flight RF05

ATTREX RF05 surveyed into the Southern Hemisphere on 9–10 March, sampling the outflow of strong convection along the South Pacific Convergence Zone (SPCZ). The flight track is shown in figure 7—the aircraft took off from Guam at 15:30 UTC on 9 March and flew a straight path southeast, reaching its furthest point away from Guam at 00:30 UTC on 10 March before

returning to Guam on a path closer to the Solomon Islands and Papua New Guinea. The aircraft returned to the vicinity of Guam at around 08:00 UTC and flew around Guam before landing at 11:00 UTC.

Large amounts of convection were present in the Southern Hemispheric portion of the Warm Pool region around the time of ATTREX RF05. A series of tropical cyclones were active at this time: tropical cyclone Gillian was in the Gulf of Carpentaria, tropical cyclone Hadi was near the east coast of Queensland, and Lusi was intensifying to become a tropical cyclone on 10

March, shown in the synoptic analysis chart in figure 8, and the MTSAT satellite imagery in figure 9.

Because the ATTREX ozone measurements are noisy, we present here average measurements taken between ∼150 hPa and the tropopause along each ascent of the aircraft. For this purpose the tropopause is defined as the point at which ozone concentrations start increasing sharply from tropospheric levels towards the stratosphere. Mean tropospheric ozone concentrations were determined for each profile by splitting the altitude range into eight equal parts in logarithmic pressure space, finding the

mean tropospheric ozone concentrations in each part, then averaging these values (omitting parts of the profile with no data from the averaging calculation). Mean concentrations so obtained from each of the profiles measured during RF05 are shown in figure 7, where it can be seen that the mean tropospheric ozone concentrations are lowest in the Southern Hemisphere, typically between 10 and 13 ppbv. These values are very similar to those measured in the TTL over Manus between 18 and 23 February. In the Northern Hemisphere, ozone concentrations on the return leg (between 06:00 UTC and 08:30 UTC on 10 March) were

∼15–16 ppbv on average, compared to the outbound leg (between 15:45 UTC and 19:45 UTC on 9 March), which were above 30 ppbv.

The relationship between the appearance of the low concentrations of ozone and areas of deep convection was investigated using the Met Office's Numerical Atmospheric-dispersion Modelling Environment (NAME) (Jones et al., 2007). NAME is a Lagrangian model in which particles are released into 3-D wind fields from the operational output of the UK Met Office Unified

Model meteorology data (Davies et al., 2005). These winds have a horizontal resolution of 17 km and 70 vertical levels, which reach ∼80 km. In addition, a random walk technique was used to model the effects of turbulence on the trajectories (Ryall et al., 2001). The NAME model was used in single-particle mode, initializing one trajectory at each point along the RF05 flight track where an ozone measurement was made. Back-trajectories were run for one day, with output at six-hour intervals.

NAME suggests that the Southern Hemisphere air originated from the southeast, in the area where tropical cyclone Lusi

was situated. Figure 9 shows the back-trajectories initialized in the troposphere (below 100 hPa) along the flight track of RF05: sections of track from which back-trajectories crossed the 800 hPa isobaric surface are shown in cyan; sections where they did not are in magenta. The yellow markers denote the final position, after 24 hours, of the back-trajectories that crossed the 800 hPa isobaric surface, indicative of rapid convection. The majority of these trajectories are in the Southern Hemispheric portion of the flight, and are projected by NAME to originate from tropical cyclone Lusi. It should be noted that the NAME



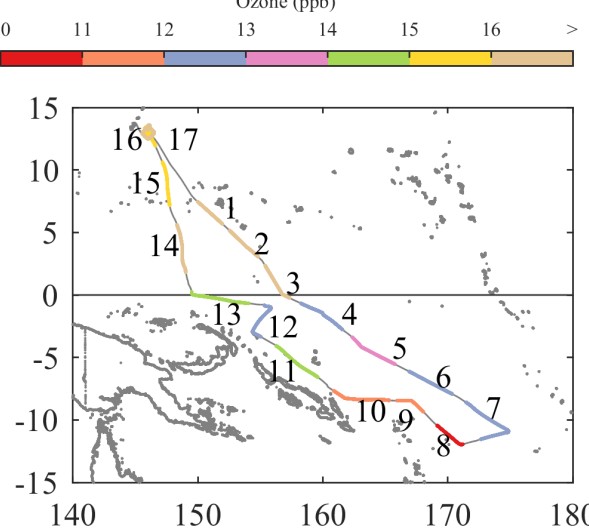

**Figure 7.** Flight track of RF05, with each profile performed by the Global Hawk chronologically numbered, and coloured by mean tropospheric ozone in each profile.

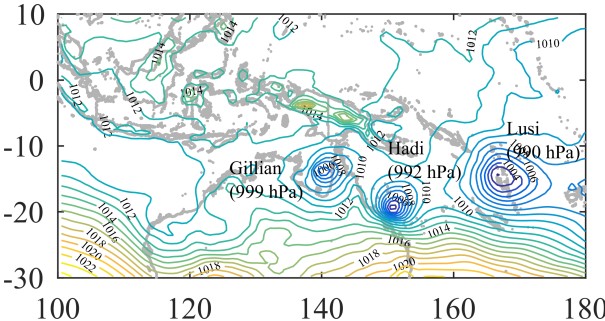

**Figure 8.** Synoptic chart from ECMWF ERA-Interim data from 10 March at 00:00 UTC. The three tropical cyclones are labelled, along with their central minimum pressure.

model cannot capture the effect of individual convective cells because of the low horizontal resolution of the meteorological data, but its convection parameterization is capable of reproducing net vertical transport over relatively large areas (Ashfold et al., 2012; Meneguz & Thomson, 2014).

The back-trajectories initialized in the Northern Hemisphere also came from the southeast, but in the twenty-four hour period of the NAME model run, the trajectories had only reached the edge of the tropical storm, so the sampled air mass passed through this region whilst the storm was developing, rather than when it was mature.

Figure 10 shows the number of back-trajectories that crossed the 800 hPa isobaric surface. None of those initialized from the Northern Hemisphere profiles 1, 2, 15, 16 and 17 crossed 800 hPa, and fewer than 1% from profiles 3, 4, 5, 13 and 14 did so. However, up to 5% of back-trajectories initialized along profiles 6–12 in the Southern Hemisphere cross the 800 hPa




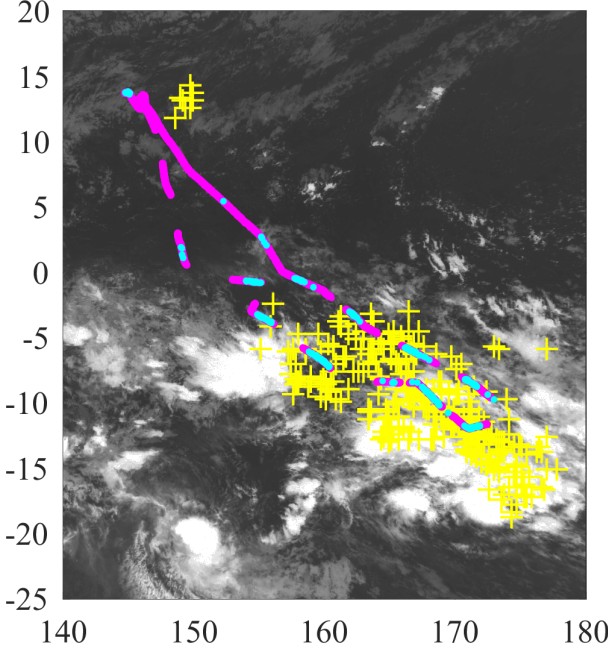

**Figure 9.** MTSAT infrared satellite image at 18:00 UTC on 8 March 2014, around 24 hours before the mid-point of flight RF05 to coincide with the endpoints of the 24 hour back-trajectories. Tropical storm Lusi is visible as the cluster of covection centred around (170°E 20°S). The tropospheric (>100 hPa) portion of the flight track of RF05 is shown in magenta, and, in the case where trajectories crossed the 800 hPa isobaric surface, in cyan. The positions after 24 hours of the trajectories that crossed the 800 hPa isobaric surface at some point in the model are marked as yellow crosses.

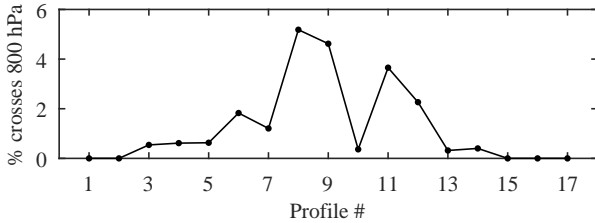

**Figure 10.** Percentage of back-trajectories crossing the 800 hPa surface within one day from the different ascent profiles on RF05.

isobaric surface, with exception of profile 10. These are also the sections with the lowest ozone concentrations, with values similar to those observed by the ozonesondes over Manus. Again, the low concentrations are consistent with recent uplift in deep convection.





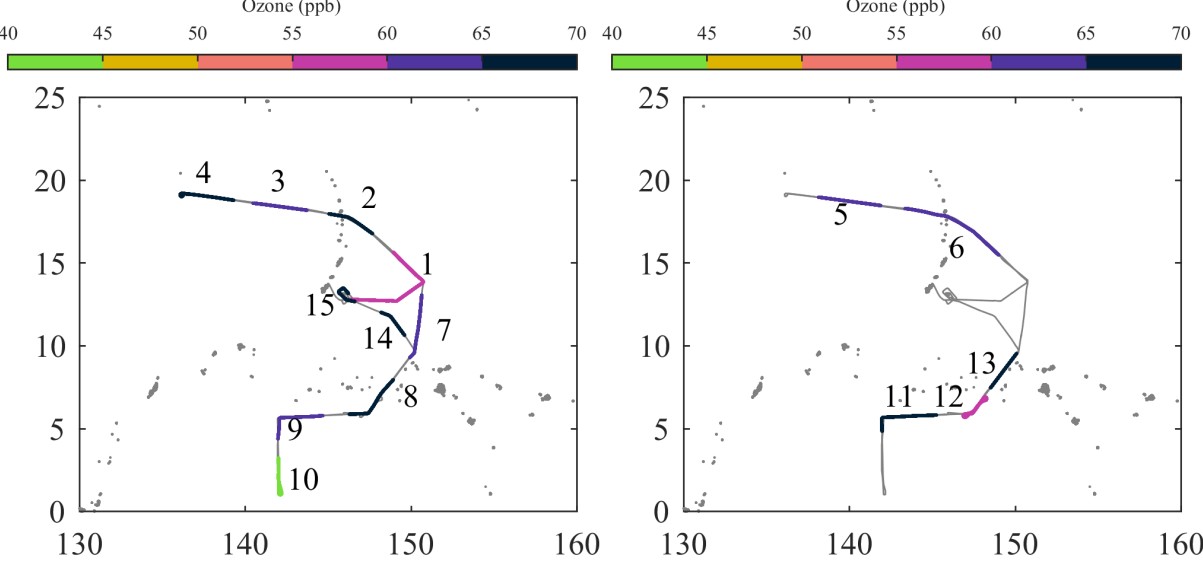

**Figure 11.** Flight track of RF01. The profiles that would otherwise be obscured by other profiles are shown on the right-hand plot.

## 4.3 Other ATTREX flights

The other research flights observed no notably low ozone concentrations in the tropical tropopause layer, hinting that the lowest ozone concentrations were confined to the Southern Hemisphere during the ATTREX campaign. RF01 flew in an arc, approximately following the streamlines of the monsoon anticyclone, which, along with most of the convection that day was

5   situated a long way to the west of Guam. Ozone concentrations on this flight were high, and none of the profiles had mean tropospheric concentrations below 30 ppbv (figure 11). Likewise, RF02, which flew within a small circle of airspace for the duration of its $17\frac{1}{2}$ hour flight providing repeated measurements of the same airmass, observed mean ozone concentrations of 25–40 ppbv (figure 12).

    RF03 flew eastwards to intercept the outflow of cyclone Faxai, before returning on a similar flight track back to Guam. Mean

10  ozone concentrations decreased below 20 ppbv on one occasion (17.5 ppbv in profile 5—figure 13), but no other examples of low ozone concentrations were observed, even in the vicinity of Faxai. RF04 flew in similar meteorological conditions as RF03, except for the dissipation of Faxai between the two flights. The flight track took the aircraft from Guam south to 6°N, where it performed a constant altitude flight along this line of latitude from 155°E to 135°E before travelling northwards and back to Guam. Similar to RF03, only one profile—16.1 ppbv in profile 1—had mean tropospheric ozone concentrations below

15  20 ppbv.

    RF06 flew north into the extra-tropics where ozone concentration are significantly higher, and is therefore not reproduced here.

    In summary, an examination of the ATTREX flight data found mean upper tropospheric ozone concentrations as low as 10 ppbv in the outflow of cyclone Lusi in the Southern Hemisphere during flight RF05, but a corresponding flight in the





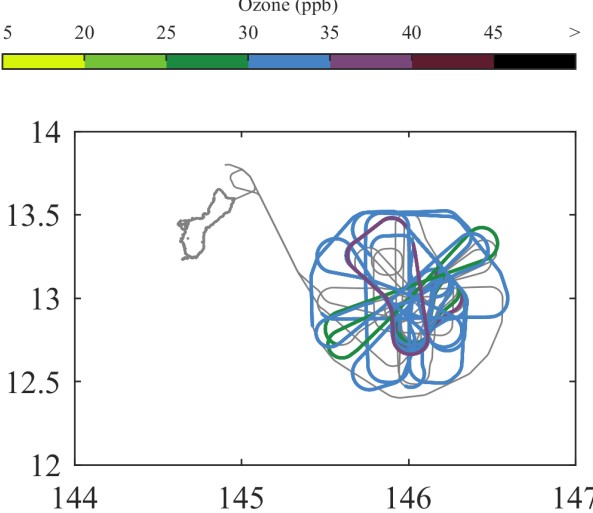

**Figure 12.** Flight track of RF02. Of the twelve profiles taken, two profiles have mean ozone concentrations of 25–30 ppbv, seven have 30–35 ppbv, and two have 35–40 ppbv.

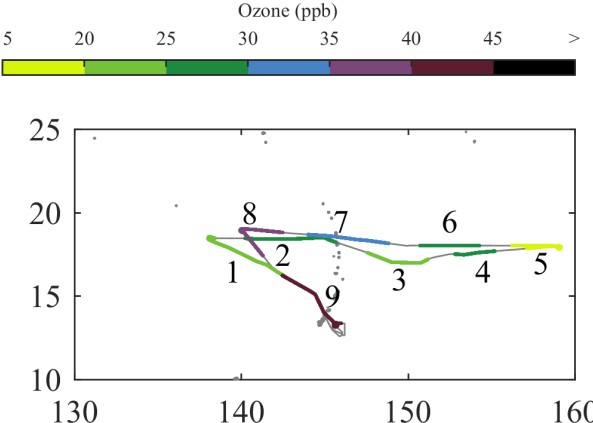

**Figure 13.** Flight track of RF03. Cyclone Faxai was situated at ∼(20°N,150°E) during this flight (see figure 5).

Northern Hemisphere in the outflow of cyclone Faxai found the lowest mean ozone concentration to be 17.5 ppbv. Meanwhile, the FAAM aircraft measured boundary layer concentrations around 10–12 ppbv between 1°S and 3°N on 4 February in a flight south from Chuuk along 152°E, values which are consistent with the Manus boundary layer measurements at that time (figure 1). Previous campaigns that measured boundary layer ozone in this region include PEM-West A of October 1991, which observed average ozone concentrations of 8–9 ppbv between the equator and 20°N (Singh et al., 1996), and PEM-tropics B in March 1999 which measured low-level ozone concentrations < 15 ppbv in the Southern Hemisphere (Browell et al., 2001; Oltmans et al., 2001), consistent with the CAST measurements. In addition, BIBLE A and B of August–October 1998 and





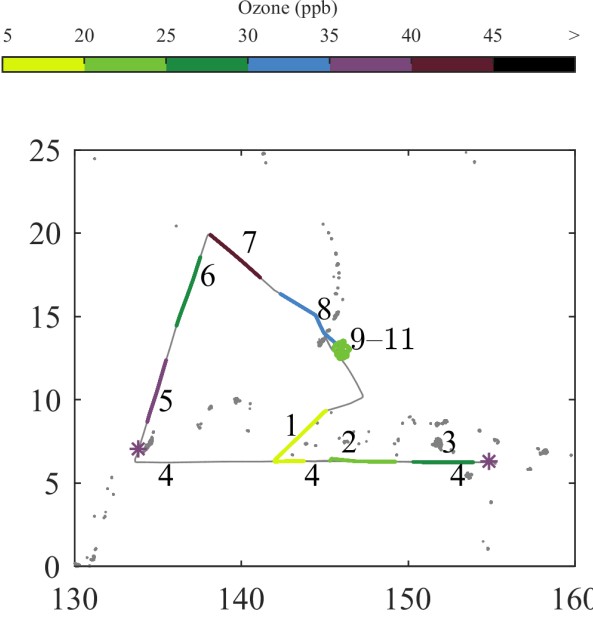

**Figure 14.** Flight track of RF04. Profile 4 contained the long, constant altitude flight, and is denoted by the start and end asterisks, to avoid obscuring the other profiles. Mean ozone in profile 4 was 38 ppbv.

1999 (Kondo et al., 2002a), measured ozone concentrations of $\sim$10 ppbv below 2 km between 2°S and 20°N (Kondo et al., 2002b).

Further evidence of an interhemispheric difference in boundary-layer ozone may be found in measurements made by the HIPPO (HIAPER Pole-to-Pole Observations) programme (Wofsy, 2011), which measured latitudinal transects of a range of trace gases along the Date Line and over the Warm Pool (see http://hippo.ucar.edu/instruments/chemistry.html). For HIPPO-1, in January 2009, and HIPPO-3, in March-April 2010 (the two missions closest in time of year to the ATTREX campaign) boundary-layer ozone concentrations below 15 ppbv were found betwen 5°N and 20°S. In addition, profiles near 15°S in January and 2°S in March-April showed values $<$ 15 ppbv extending up to 5 km. By contrast, ozone concentrations in January were $>$15 ppbv north of 5°N in the boundary layer and $>$20 ppbv north of 5°S above 2 km, while in March-April they exceeded 20 ppbv at all altitudes north of 1°N. It is therefore likely that the differences measured in the TTL in ATTREX originated from the inter-hemispheric differences in boundary layer ozone concentrations.

## 5   CAST and CONTRAST ozone measurements

The ozonesonde and Global Hawk measurements found TTL concentrations below 15 ppbv only in the Southern Hemisphere, but the sparse sampling means that similar layers in the Northern Hemisphere may just have been missed. Many more flights of the CAST and CONTRAST aircraft were made during February 2014, extending from sea level to 120 hPa. We now examine the measurements from these aircraft for evidence of very low ozone concentrations.





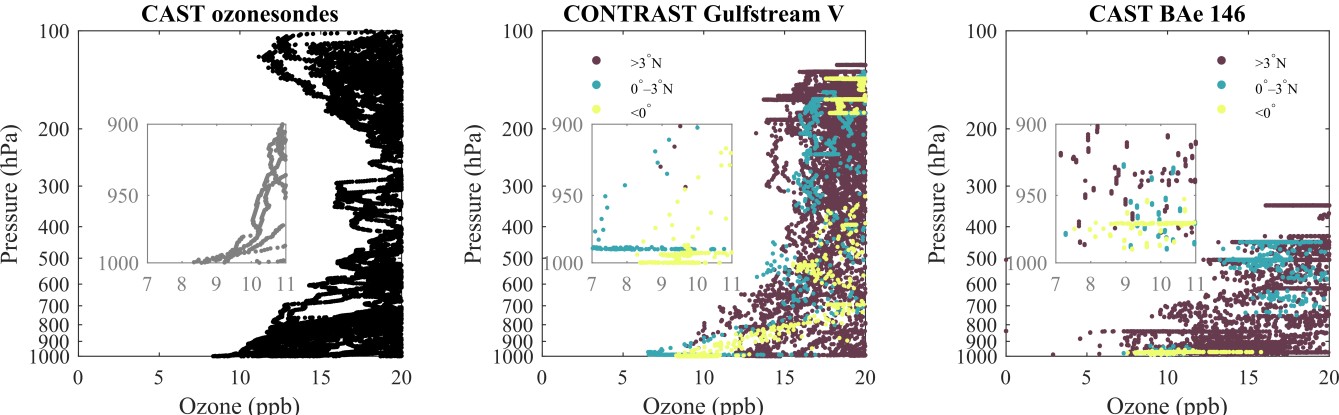

**Figure 15.** Complete dataset of ozone measurements from the CAST ozonesondes (left), the CONTRAST Gulfstream V aircraft (centre) and the CAST FAAM BAe 146 aircraft (right), with the aircraft data split into the Southern Hemisphere measurements (yellow), equator–3°N (blue) and higher Northern Hemisphere latitudes (purple). In all cases, minimum ozone near the surface was lower than minimum ozone in the mid-troposphere. The insets show the low ozone concentrations measured in the lowest 100 hPa of the atmosphere in each case.

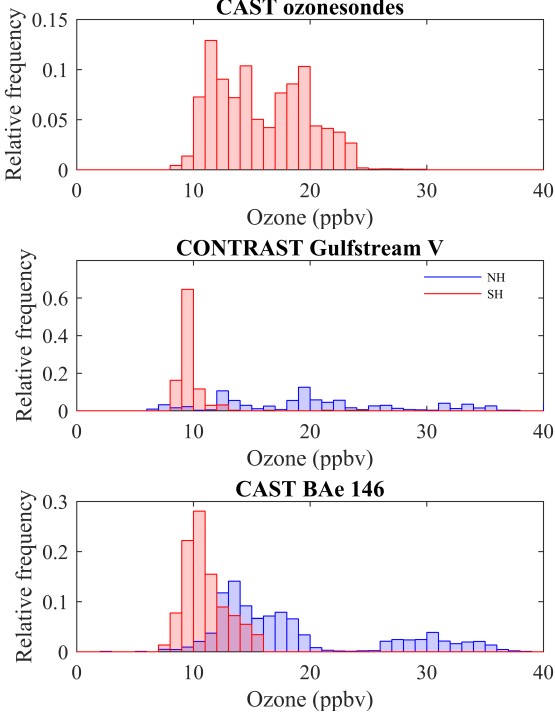

**Figure 16.** Histograms of ozone concentrations in the lowest 100 hPa from CAST ozonesondes, CONTRAST Gulfstream V and the CAST FAAM BAe 146 aircraft, split into Northern Hemisphere data (blue) and Southern Hemisphere data (red).



The FAAM BAe 146 aircraft focused on measuring close to the surface and within the boundary layer, making twenty-five flights between 18 January and 18 February (Harris et al., 2017). The NCAR Gulfstream V mostly measured in the upper troposphere in the region of main convective outflow, although many measurements were also made in the boundary layer (Pan et al., 2017); it conducted thirteen research flights and three transit flights between 11 January and 28 February.

Other than the brief excursion to 1°S on 4 February, the FAAM aircraft flew exclusively in the Northern Hemisphere, as its range was insufficient to reach the Southern Hemisphere from Guam. The NCAR Gulfstream V crossed into the Southern Hemisphere on only two occasions, thus its measurements also are predominantly in the Northern Hemisphere.

The ozone data from the two aircraft and the CAST ozonesondes from Manus Island (see section 3) are summarized in figure 15. In all three panels, ozone concentrations below 15 ppbv are frequently found below 700 hPa. The ozonesondes show

a second region of ozone-poor air in the TTL, corresponding to the event discussed in Section 3, but the CONTRAST data do not—there are isolated examples around 15 ppbv up to ~180 hPa but not above. The FAAM aircraft's ceiling is around 300 hPa so it did not measure in the TTL, but again there are only a few ozone values lower than 15 ppbv above 700 hPa. The histograms of the boundary layer measurements by the CAST ozonesondes and the CAST and CONTRAST aircraft (figure 16) show the majority of the measurements in the Southern Hemisphere were below 15 ppbv, whereas the Northern Hemisphere

measurements were broadly distributed with a large proportion of measurements above 20 ppbv.

These results are consistent with a hemispheric difference in the ozone distribution in the boundary layer over the Warm Pool, which resulted in the layers of very low ozone concentrations being lifted to the TTL around the equator and further south, but not in the Northern Hemisphere.

## 6   Very short lived substances

Whole air samplers (WAS) on board the CAST, CONTRAST and ATTREX aircraft provided measurements of very short lived substances (VSLSs), of which eight were measured by all three aircraft: dimethyl sulfide ($(CH_3)_2S$), iodomethane ($CH_3I$), tribromomethane ($CHBr_3$), dibromochloromethane ($CHBr_2Cl$), bromochloromethane ($CH_2BrCl$), dichloromethane ($CH_2Cl_2$), dibromomethane ($CH_2Br_2$) and trichloromethane ($CHCl_3$). Dichloromethane and trichloromethane are industrially produced chemicals with a strong anthropogenic signal, and are thus not plotted here; these, along with the other species measured by

the CONTRAST and ATTREX aircraft but not by CAST are plotted in the supplementary material.

The atmospheric lifetimes of these molecules range from a few minutes to a few months: $(CH_3)_2S$ has a lifetime between 11 minutes and 46 hours (Marandino et al., 2013); $CH_3I$ a lifetime of ~4 days and $CHBr_3$ a lifetime of ~15 days (Carpenter et al., 2014); that of $CHBr_2Cl$ is about three months, and $CH_2BrCl$, $CH_2Br_2$ and $CH_2Cl_2$ have lifetimes of the order of six months (Montzka et al., 2010; Mellouki et al., 1992; Leedham Elvidge et al., 2015; Khalil & Rasmussen, 1999).

All six compounds have significant marine sources. $(CH_3)_2S$, $CHBr_3$ and $CH_2Br_2$ are produced by phytoplankton (Dacey & Wakeham, 1986; Quack et al., 2007; Stemmler et al., 2015). $CH_3I$ is produced by cyanobacteria and picoplankton (Smythe-Wright et al., 2006) and large concentrations of $CH_3I$ are present in the marine boundary layer (Maloney et al., 2001). $CHBr_2Cl$





**Figure 17.** Panel plot of six compounds measured by the whole air samplers on board the three aircraft. The red line indicates the average profile of measurements taken where ozone was above 20 ppbv, and the blue line is the average profile for ozone below 20 ppbv.

is produced naturally by various marine macroalgae (Gschwend et al., 1985), and $CH_2BrCl$ is emitted by tropical seaweed (Mithoo-Singh et al., 2017).

The vertical profiles obtained from combining all of the whole air sample data from the entire CAST, CONTRAST and AT-TREX campaigns yield the plots in figure 17. Necessarily the vast majority of these points are from the Northern Hemisphere,

5   away from the very low ozone concentrations described in figure 4.2. Each data point is coloured by the ozone concentration measured at the time the WAS bottles were being filled, and average profiles for each compound were obtained for WAS samples with ozone concentrations less than 20 ppbv (in blue), and for WAS samples with ozone concentrations greater



than 20 ppbv (in red). The average profiles were generated by binning the data into twenty equally sized bins in logarithmic pressure-space between 1000 hPa and 10 hPa, and averaging the data within that bin.

$(CH_3)_2S$, $CH_3I$, $CHBr_3$ and $CHBr_2Cl$ all show enhancements when ozone concentrations were less than 20 ppbv compared to when ozone concentrations were greater than 20 ppbv, suggesting that the more ozone-deficient air has encountered the

marine boundary layer, where these molecules are produced, more recently than the more ozone-rich air. Meanwhile, $CH_2BrCl$ and $CH_2Cl_2$ show no difference when ozone is low and when ozone is high because they have a much longer lifetime than the other VSLSs measured, and $CH_2Cl_2$ also has large industrial emissions.

The remaining species that were measured with the whole air samplers from CONTRAST and ATTREX are plotted in the supplementary material. These thirty-three molecules show expected enhancements in the typically polluted high-ozone régime

for those of industrial origin, and enhancements in the typically cleaner low-ozone régime for those of marine origin.

Very few WAS samples were taken in the Southern Hemisphere—only two were taken by the FAAM BAe 146, 60 by the Global Hawk, and 134 by the Gulfstream V, compared to the Northern Hemisphere where 302 FAAM samples, 1373 Gulfstream V samples and 608 Global Hawk samples were taken. As a result, very few WAS samples were taken in areas where ozone concentrations were at their lowest during the campaign, and further investigation of these halomethanes during

very low ozone events would be beneficial in future campaigns.

## 7    Conclusions

We have presented an extensive dataset of ozone observations from three research aircraft and ozonesondes over the West Pacific Warm Pool in February-March 2014, with a particular focus on the TTL. The results point to the generation of layers with very low ozone concentration (< 15 ppbv) just below the tropopause due to uplift by deep convection, confirming the

conclusion of Newton et al. (2016) based on the ozonesonde data. The lowest values measured in the TTL, around 10–12 ppbv, are very similar to those measured in the boundary layer in the region, consistent with uplift of boundary-layer air up to the tropopause region. This places boundary-layer air above the level of net radiative heating in the TTL and therefore in a position to ascend into the stratosphere in the Brewer-Dobson circulation. Consequently, it provides a route for very short-lived halocarbon species to reach the stratosphere. Evidence from the extensive whole air samplers carried by the three aircraft shows

a negative correlation between ozone and species of marine origin, consistent with uplift in convection.

Despite the far more extensive sampling of the Northern Hemisphere than the Southern during the aircraft campaign, very low ozone concentrations in the TTL were only found in the Southern Hemisphere; even in the outflow of Cyclone Faxai the Global Hawk measured 15 ppbv of ozone, similar to measurements in convective anvils by the Gulfstream V in the Northern Hemisphere. This suggests a hemispheric difference in the TTL ozone distribution, either because of lower boundary-layer

ozone concentrations in the Southern Hemisphere or because of differences in the convective uplift. Previous measurement campaigns in this region point to an interhemispheric difference in boundary-layer ozone concentration as being responsible for the corresponding feature in the TTL.



*Acknowledgements.* We thank the NERC Facility for Airborne Atmospheric Measurements (FAAM) for the CAST aircraft data, and the Centre for Data Archival (CEDA) for supporting meteorological data. The project was supported by Natural Environment Research Council (NERC) grant NE/J006173/1. Richard Newton is a NERC-supported research student.



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
