# Peer review of "Observations of ozone-poor air in the Tropical Tropopause Layer"

_Atmospheric Chemistry and Physics, 2017_

## Referee Comment (RC1) · Anonymous Referee #1 · 13 Nov 2017

Newton, Vaughan, et al: Observations of Ozone-Poor Air in the Tropical Tropopause Layer [TTL]

The essence of the paper is that by combining aircraft and sounding observations from Feb-March 2014, it can be shown that SH (southern hemisphere) ozone in and below the TTL over the TWP (tropical western Pacific) is lower (approx. half) the concentrations over the corresponding NH (northern hemisphere). Evidence is presented for the lower ozone in the SH TTL (less than 10 ppbv) as due to deep convection of low-ozone air in the marine boundary layer below 300 m. Above that level to about 200 hPa ozone is not less than 10 ppbv in either NH or SH. Although limited in the amount of SH data, reactive halogenated hydrocarbons of biogenic origin and sampled by aircraft, are suitable tracers for implicating deep convection as the mechanism for transporting

the low-ozone air from surface to TTL. The study is of interest and should be published with modifications that acknowledge a number of other papers that describe NH-SH ozone gradients (from surface and aircraft) and/or very low ozone in the upper troposphere and TTL throughout the tropical Pacific. The latter include PEM-Tropics (1990s) and TC4 (2007). In other words, the CAST-CONTRAST-ATTREX analysis is illuminating but does not show any surprises. The very important, motivating question posed on page 2, namely, "do enough reactive species penetrate the lower stratosphere to perturb the composition?" is not really answered. It is recommended that the authors conclude the paper with summarizing how their findings address this issue.

Relevant papers:

Avery, M. A., et al: Convective distribution of tropospheric ozone and tracers in the central American ITCZ Region: Evidence from observations during TC4, J. Geophys. Res., 115, D00J21, doi: 10.1029/2009JD013450, 2010. Petropavlovskikh, I., et al: Low ozone bubbles observed in the tropical tropopause layer during the TC4 campaign in 2007, J. Geophys. Res.,115, D00J16, doi: 10.1029/2009JD012804, 2010. Pickering, K. E., et al: Trace gas transport and scavenging in PEM-Tropics-B SPCZ convection, J. Geophys. Res., 106, 32591-32608, 2001. Thompson, A. M., et al: Convective and wave signatures in ozone profiles over the equatorial Americas: Views from TC4 (2007) and SHADOZ, J. Geophys. Res., 115, D00J23, doi: 10.1029/2009JD012909, 2010.

---

## Referee Comment (RC2) · Anonymous Referee #2 · 6 Dec 2017

General comments:

This paper addresses core elements of the hypothesis, originally proposed by Kley et al. (1996) and subsequently addressed by Folkins et al. (2002) and others, that low values of ozone in the TTL over the Pacific Warm Pool are a consequence of the convective transport of ozone-depleted air from the boundary layer over the region. To address this hypothesis, the paper employs data the coordinated airborne measurements from the CAST, CONTRAST and ATTREX airborne campaigns staged from Guam in early 2014, together with a concurrent CAST equatorial ozonesonde campaign at Manus Island. The analysis presents three significant findings: (a) that very low ozone (<15 ppbv) occurs at least episodically in the Warm Pool boundary layer, (b) that similarly low ozone values occur in the TTL south of the equator over the tropical Western Pacific but only rarely north of the equator, and (c) an association of of dimethyl sulfide, methyl iodide and other very short-lived substances (VSLS) of marine origin with lower ozone in the upper troposphere. While findings (a) and (c) are reasonably robust, (b) suffers from an unknown level of uncertainty due to the high degree of noise in the ATTREX ozone data which are the basis of the finding. This is a major shortcoming of the paper.

Scientific significance:

Taken together the main findings are not inconsistent with the original hypothesis. However, due to the operational constraints on the three aircraft, based as they were at 13°N, relatively few measurements were obtained in the Southern Hemisphere, where deep convection was presumably providing the hypothesized direct path of low-ozone air in the boundary layer to the TTL. The NASA Global Hawk used in ATTREX did have the range and duration to sample effectively south of the equator from Guam, however it only did so on one occasion, and this was after the CAST and CONTRAST campaigns had concluded.

Despite the limitations of the sampling, the finding of a hemispheric contrast in TTL ozone is significant. As the authors state in their conclusions, this contrast may be a result of differences in the ozone content of the air entering convection in the two hemispheres or a difference in convection. They favor the latter as it is consistent with seasonal boundary layer measurements from earlier campaigns, including from GTE, BIBLE and HIPPPO.

There is one important shortcoming in the analysis however which renders the finding of hemispheric contrast as presented less than robust; this is the high level of noise in the Global Hawk UCATS ozone data. As I discuss below, this noisiness and its impact on the uncertainty in the analysis needs to be addressed directly in the paper.

The second finding is the association of trace constituents of marine origin with lowered ozone in the upper troposphere. I suspect that if more observations had been obtained

from the active Hadley branch south of the Equator, the association would have stood out much more clearly. Nevertheless, it is an important finding as it stands. Maybe not a 'smoking gun', but tantalizing nonetheless.

Scientific quality:

The paper provides a somewhat limited review of the scientific literature of the TTL low-ozone problem. In its simplest form, the hypothesis is straightforward. However, testing the hypothesis is made difficult for a number of reasons, not the least of which is the challenge of sampling a phenomenon so vast in scale. A discussion of this and the challenges that it presented to the utilization of the disparate datasets in the paper would have been as important as the review of the basic elements of the phenomenon. Thus, the paper would benefit from a more focussed detailing of what they are looking for in the various analyses they lay out and what the train of the argument will be. While this becomes clearer as you go along in the text, it would have been better if the reasoning had been presented at the outset.

There are four sections of analysis focusing respectively on the CAST/Manus ozonesonde profiles (Section 3), the TTL ozone measurements on the Global Hawk (Section 4), the CAST and CONTRAST ozone measurements (Section 5) and the very short-lived substances measured by the Whole Air Samplers on all three aircraft (Section 6). In Section 3, a time-height section of ozonesonde profiles is presented that clearly shows the presence of sub-15 ppbv ozone in both the TTL and the boundary at this south equatorial location, though the low ozone in the TTL was not linked in any way with the local boundary layer values. Indeed, the HYSPLIT trajectory results, which place the origin of the latter to the east of Manus, illuminate the challenge of making inferences from a single location's data. While the authors don't say this in so many words, they clearly wanted to set the stage here for their subsequent analysis of the ATTREX Global Hawk TTL measurements.

Indeed, Section 4, which focuses on those measurements, comprises the bulk of the

paper, and of this, most is devoted to the results from the single ATTREX flight (RF05) that sampled the Southern Hemisphere. The key figure here is Fig. 7 which plots the average TTL ozone along extended sections of the RF05 track. This averaged presentation is an accommodation to the fact that the Global Hawk ozone data were extremely noisy. As is stated in the paper, ozone data were plotted as averages in each slow ascent of the aircraft from 150 hPa to the tropopause; left unexplained is why the ascent segments appear in the figure to be coterminous. (Are the descents nearly instantaneous?) In any event, if the ozone data had been of higher quality, then it would have been helpful to augment the flight track figure with a standard line plot.

Given that the UCATS ozone data are so central to the analysis in this section, indeed the whole paper, the rather severe shortcomings of UCATS ozone measurement mentioned in passing in Section 4 ought to have been discussed fully in Section 2 on instrumentation. Particularly concerning is the possible negative bias of up to 5 ppbv at low ozone concentrations. Given the possibility of such a substantial bias, the reader might fairly ask how much confidence can be placed in the hemispheric difference suggested in Fig. 7. I would suggest that the authors show at least an extended section of the ozone data in time series format to give the reader a better sense of the uncertainty in the averaged values.

In contrast to Section 4, the subsequent section is compelling, showing striking similarities between the Manus ozonesonde data and the ozone data on board the CON-TRAST Gulfstream V and the CAST BAe 146, the rather different regions of sampling notwithstanding. Notably absent in this section are the ATTREX ozone data, which but for their extreme noisiness would have filled in the region above the ceiling of the Gulfstream V. So the picture is unfortunately incomplete.

Section 6 shows in Figure 17 some separation between profiles of Very Short-lived Substances originating in the marine boundary layer on the basis of ozone. Here again, though, the noisiness of the UCATS ozone data cloud the picture in the critical TTL altitudes. By lumping all the WAS data together, the differences in the quality of

the ozone data between aircraft are essentially blurred. Would, for example, we see the same strong difference in methyl iodide at 300 hPa if each aircraft were plotted separately? Indeed, where do the Global Hawk data leave off? Are Global Hawk descent profiles at Guam used at all in Figure 17? Here again then, as in Section 5, questions about the quality of the UCATS ozone data limit the confidence in the authors' conclusions.

Presentation quality:

This paper is very well written, and is almost devoid of grammatical errors. The figures, with the exception of Figure 7 noted above, are well conceived and competently drafted.

Recommendation:

I recommend that the paper be accepted with major revisions. The most important of these would be a thorough analysis of the UCATS ozone data quality and the uncertainties in the averaging they base their central argument upon. I would also suggest that the WAS data from all three aircraft not be combined as they are in Figure 17 unless they can present evidence that there aren't biases between the ozone measurements on board the three aircraft.

With regard to the flow of the text itself, I would recommend a more detailed background section in the introduction. In particular, I think that the authors should provide the reader with some sense of what questions can be robustly addressed by the datasets acquired in this unprecedented joint effort - and what questions are likely to be left unanswered. With that greater clarity, this paper may well stand as an important contribution.

---

## Author Comment (AC1) · 17 Feb 2018

We thank the reviewer for their comments on the paper. We respond below to the specific comments made.

1. *[The authors] should acknowledge a number of other papers that describe NH-SH ozone gradients (from surface and aircraft) and/or very low ozone in the upper troposphere and TTL throughout the tropical Pacific. The latter includes PEM-Tropics (1990s) and TC4 (2007).*
   The introduction has been expanded to include discussion of TC4. The PEM series of flights only measured up to 12 km [Hoell et al., 1999; Raper et al., 2001] and are not relevant to a discussion of the TTL. We refer to PEM-West and PEM-

tropics in section 4.4 of the paper where we discuss boundary-layer ozone. The following passage has been added to the introduction:

> Bubbles of relatively low ozone have also been observed in other parts of the world. During the TC4 campaign, anomalously low ozone concentrations of ∼60 ppbv were found at 14–16 km altitude in the TTL off the coast of Ecuador—typical values of ozone at this altitude in this region were measured to be ≥100 ppbv. These low-ozone bubbles were also shown to be a result of non-local convection followed by advection to where it was measured by the NASA DC-8 aircraft [Petropavlovskikh et al., 2010].

2. *The very important motivating question posed on page 2, namely, "do enough reactive species penetrate the lower stratosphere to perturb the composition?" is not really answered. It is recommended that the authors conclude the paper with summarizing how their findings address this issue.*

   The focus of this paper, and the objective described on page 2, was to "corroborate the ozonesonde measurements with other measurements and to determine how widespread these bubbles of low-ozone air over the Warm Pool," rather than whether reactive species penetrate into the lower stratosphere. The key question posed in the introduction was "whether deep convection is nevertheless capable of lifting very short-lived halogenated species near enough to the tropopause that their breakdown products reach the stratosphere" rather than direct penetration of the lower stratosphere. We don't think that the reviewer has read this part of the paper correctly and therefore do not make any changes to the paper.
**References**

Hoell, J. M., Davis, D. D., Jacob, D. J., Rodgers, M. O., Newell, R. E., Fuelberg, H. E., McNeal, R. J., Raper, J. L., Bendura, R. J. (1999). Pacific Exploratory Mission in the tropical Pacific: PEM-Tropics A, August–September 1996. *Journal of Geophysical Research: Atmospheres*, 104(D5), 5567–5583.

Petropavlovskikh, I., Ray, E., Davis, S. M., Rosenlof, K., Manney, G., Shetter, R., Hall, S. R., Ullmann, K., Pfister, L., Hair, J., Fenn, M., Avery, M., Thompson, A. M. (2010). Low-ozone bubbles observed in the tropical tropopause layer during the TC4 campaign in 2007. *Journal of Geophysical Research: Atmospheres*, 115(D10).

Raper, J. L., Kleb, M. M., Jacob, D. J., Davis, D. D., Newell, R. E., Fuelberg, H. E., Bendura, R. J., Hoell, J. M., McNeal, R. J. (2001). Pacific Exploratory Mission in the Tropical Pacific: PEM-Tropics B, March-April 1999. *Journal of Geophysical Research: Atmospheres*, 106(D23), 32401–32425.

---

## Author Comment (AC2) · 17 Feb 2018

We thank the reviewer for their careful and considered comments. We respond below to the specific comments made.

1. *The paper provides a somewhat limited review of the scientific literature of the TTL low-ozone problem.*
   A similar comment was provided by Anonymous Referee #1, and is addressed in the Response to Anonymous Referee #1.

2. *The paper would benefit from a more focussed detailing of what they are looking for in the various analyses they lay out and what the train of the argument will be. While this becomes clearer as you go along in the text, it would have been better*

[Figure]

*if the reasoning had been presented at the outset.*

A new section in the introduction, labelled "Article overview" has been created to do this. The following text has been added to this section:

> In section 2 we describe the instruments that were used on board the three aircraft to collect the measurements described in this article. Section 3 provides a brief overview of the CAST ozonesonde measurements from Manus, which were described in detail in Newton et al. [2016], that provided the first evidence of the occurrence of localized low ozone concentrations during the campaign.
>
> We then introduce the Global Hawk ozone profiles in section 4, concentrating on one flight that sampled well into the Southern Hemisphere from Guam in section 4.2—this flight produced further evidence of low-ozone concentrations, especially in the Southern Hemisphere portion of the flight. Within this section, we also discuss the uncertainties, and implications thereof, of the UCATS ozone instrument on board the Global Hawk, and how we approached the issue of noisiness in the UCATS dataset. This is followed by a brief discussion of the other AT-TREX flights in section 4.3.
>
> Section 5 discusses the lower troposphere measurements that were made by the CAST and CONTRAST aircraft, providing information on boundary layer ozone concentrations that can be used to infer the origin of low ozone in the TTL. Section 6 shows a subset of the very short lived substances (VSLS) that were measured using Whole Air Samplers (WAS) on board all three aircraft, showing the composition differences between the VSLSs in low-ozone and high-ozone cases to infer that recently convected ozone-deficient air has a distinct chemical composition compared to high-ozone cases. (A supplementary section contains the full dataset of WAS VSLS chemical data.) Finally section

7 summarizes the findings of this article.

3. *Left unexplained is why the ascent segments appear in the figure to be cotermi-nous. (Are the descents nearly instantaneous?)*
Changes made to section 4 clarify this diagram, which has been modified to show exactly where the aircraft was below the tropopause. The following text has been added to the caption of figure 7 to ensure this confusion is not encountered by other readers.

The grey line shows the flight path between profiles.

4. *Given that the UCATS ozone data are so central to the analysis in this section, indeed the whole paper, the rather severe shortcomings of UCATS ozone mea-surement mentioned in passing in Section 4 ought to have been discussed fully in Section 2 on instrumentation. Particularly concerning is the possible negative bias of up to 5 ppbv at low ozone concentrations. Given the possibility of such a substantial bias, the reader might fairly ask how much confidence can be placed in the hemispheric difference suggested in Fig. 7. I would suggest that the au-thors show at least an extended section of the ozone data in time series format to give the reader a better sense of the uncertainty in the averaged values.*
We thank the referee for this comment which did indeed identify a serious omis-sion in the original paper. We hope this is now rectified, in the form of a new section 4.2 discussing systematic errors in the UCATS data and an enlarged section 4.3 (old 4.2) discussing the random errors and why we choose the ascent sections of the flights rather than both ascent and descent in the colour line plots. Fig. 11 shows a time series of (averaged) data with error bars along RF05.

5. *Section 6 shows in Figure 17 some separation between profiles of Very Short-lived Substances originating in the marine boundary layer on the basis of ozone. Here again, though, the noisiness of the UCATS ozone data cloud the picture in the critical TTL altitudes. By lumping all the WAS data together, the differences in*

*the quality of the ozone data between aircraft are essentially blurred. Would, for example, we see the same strong difference in methyl iodide at 300 hPa if each aircraft were plotted separately? Indeed, where do the Global Hawk data leave off? Are Global Hawk descent profiles at Guam used at all in Figure 17? Here again then, as in Section 5, questions about the quality of the UCATS ozone data limit the confidence in the authors' conclusions.*

A new section in the supplementary material has been created to address this entitled "Principal WAS chemicals split by aircraft", which shows three panel plots similar to figure 17 (figures S1–S3) but with the data from just one aircraft plotted on each. The conclusion from these plots was that there was little data from the CAST aircraft compared to CONTRAST and ATTREX, so this dataset would not have much of an effect on the overall averages, and the overlap between the CONTRAST and ATTREX data occurs only between 150 hPa and 180 hPa. The overall trends in the six principal WAS chemicals hold true when plotting only the CONTRAST data and when plotting only the ATTREX data.

**References**

Newton, R., Vaughan, G., Ricketts, H. M. A., Pan, L. L., Weinheimer, A. J., Chemel, C. (2016). Ozonesonde profiles from the West Pacific Warm Pool: measurements and validation. *Atmospheric Chemistry and Physics*, 16(2), 619–634.

---

## Author Response (AR1)

**Observations of ozone-poor air in the Tropical Tropopause Layer**

[revised manuscript text omitted]

**1.2   Article overview**

In section 2 we describe the instruments that were used on board the three aircraft to collect the measurements described in this article. Section 3 provides a brief overview of the CAST ozonesonde measurements from Manus, which were described in detail in Newton et al. (2016), that provided the first evidence of the occurrence of localized low ozone concentrations during the campaign.

We then introduce the Global Hawk ozone profiles in section 4, concentrating on one flight that sampled well into the Southern Hemisphere from Guam in section 4.3—this flight produced further evidence of low-ozone concentrations, especially in the Southern Hemisphere portion of the flight. Within this section, we also discuss the uncertainties, and implications thereof, of the UCATS ozone instrument on board the Global Hawk, and how we approached the issue of noisiness in the UCATS dataset. This is followed by a brief discussion of the other ATTREX flights in section 4.4.

Section 5 discusses the lower troposphere measurements that were made by the CAST and CONTRAST aircraft, providing information on boundary layer ozone concentrations that can be used to infer the origin of low ozone in the TTL. Section 6 shows a subset of the very short lived substances (VSLS) that were measured using Whole Air Samplers (WAS) on board all three aircraft, showing the composition differences between the VSLSs in low-ozone and high-ozone cases to infer that recently convected ozone-deficient air has a distinct chemical composition compared to high-ozone cases. (A supplementary section contains the full dataset of WAS VSLS chemical data.) Finally section 7 summarizes the findings of this article.

[revised manuscript text omitted]
 transfer flight from Armstrong Flight Research Center in California to Andersen Air Force Base in Guam on 16 January and the return flight on 13 March made few measurements in the West Pacific region and are not considered here.

Flight RF01 on 12–13 February focused on the composition, humidity, clouds and thermal structure of the Northern Hemisphere part of the Warm Pool region. Convection was situated mostly around the Maritime Continent on this day (figure 3), with no notable convection around Guam. The second flight, RF02 occurred on 16–17 February with similar scientific objectives to RF01. As a result of a satellite communications problem, the aircraft was required to stay in line-of-sight contact with the airbase in Guam, and consequently the aircraft flew in a small area of airspace close to

[Figure]

**Figure 3.** MTSAT infrared satellite image from 12 February at 12:00 UTC, coincident with flight RF01 (yellow track). Green asterisk denotes location of Guam; magenta asterisk that of Manus Island. Convection is centred mostly around the Maritime Continent on this day.

[Figure]

**Figure 5.** Satellite image of March 4 at 12:00 UTC, coincident with flight RF03. Cyclone Faxai is visible to the northeast of Guam.

[Figure]

**Figure 4.** As figure 3 but for 16 February at 12:00 UTC coincident with flight RF02. A band of convective activity is visible to the southeast of Guam.

[Figure]

**Figure 6.** Satellite image of March 6 at 12:00 UTC, coincident with flight RF04. Convection is minimal in the Northern Hemisphere, and is concentrated mostly in the Southern Hemisphere.

the island. On this day, convection was visible to the southeast of Guam in the MTSAT satellite imagery (figure 4).

The third flight took place after a two-week hiatus on 4–5 March. Its objectives were to sample the outflow of tropical cyclone Faxai, which developed in the region in the previous few days, with vertical profiles performed to observe the outflow cirrus cloud from the cyclone. Apart from tropical cyclone Faxai, the majority of the convection was in the Southern Hemisphere around Papua New Guinea (figure 5).

Flight RF04 took place on 6–7 March. Tropical cyclone Faxai had dissipated by this time, leaving a dearth of convection in the Northern Hemisphere; the most convectively active region was around Papua New Guinea (figure 6). RF05 surveyed the Southern Hemisphere on 9–10 March, measuring the lowest ozone concentrations observed by the Global Hawk during the ATTREX campaign. This flight is discussed in detail in the next section. The final research flight, RF06, took place on 11–12 March, surveying latitudes north of 10°N either side of the subtropical jet, and is outside the scope of this paper. A full description of the ATTREX flights and meteorological conditions encountered can be found in Jensen et al. (2017).

**4.2    Systematic errors in ATTREX ozone data**

As discussed in section 2, previous studies have suggested there may be a low bias in UCATS ozone measurements. Fortunately, flight RF02 on 16/17 February offered an opportunity to examine data from this campaign for any evidence of such a bias. As shown in figure 4 (and later, figure 16), the entire flight took place just south-east of Guam, within an area spanning 1° in latitude and longitude.

[Figure]

**Figure 7.** Intercomparison of UCATS descent ozone profile on RF02 with nearby profiles from the Gulfstream V and an ozonesonde. The Global Hawk profile was measured between 09:20 and 10:45 UTC, the sonde between 08:46 and 10:44 UTC and the Gulfstream V between 05:02 and 05:23 UTC.

Figure 7 shows a comparison between UCATS data measured on the descent to Guam, the descent profile from the Gulfstream V (around 5 hours earlier), and the descent profile from an ozonesonde coincident in time with the Global Hawk descent. (The ozonesondes were flown with a valved balloon, allowing both ascent and descent rates to remain below 6 m s$^{-1}$, thus enabling ozone measurements on descent). The UCATS ozone follows the descent sonde profile closely

below 10 km, but in the region of lowest ozone concentration, between 12 and 14 km, it tends to be more consistent with the Gulfstream V, although showing rather more structure than the other two profiles.

Given the obvious variability in tropospheric ozone shown by figure 7, a quantitative validation of the UCATS ozone would require many more comparisons. We can conclude however that there is no evidence from this example of a low bias in the UCATS measurements at low concentrations. This allows measurements across the Tropical Warm Pool by the Global Hawk to be used to explore regions of very low ozone concentration.

**4.3    ATTREX flight RF05**

ATTREX RF05 surveyed into the Southern Hemisphere on 9–10 March, sampling the outflow of strong convection along the South Pacific Convergence Zone (SPCZ). The aircraft took off at 15:30 UTC on 9 March and flew a straight path southeast, reaching its furthest point away from Guam at 00:30 UTC on 10 March before returning on a path closer to the Solomon Islands and Papua New Guinea. The aircraft returned to the vicinity of Guam at around 08:00 UTC and flew around the island before landing at 11:00 UTC. Figure 8 shows the altitude of the aircraft during RF05: after the initial ascent there were basically 18 repeats of a relatively rapid descent, a short level section near 14.5 km, and relatively slow ascent (with the exceptions that there was no level section in set 10 and the aircraft descended to Guam after the final level section).

[Figure]

**Figure 8.** Altitude of Global Hawk on RF05 as a function of time. The flight has been divided into sections, numbered at the top of the plot; these numbers correspond to the section or profile numbers in figures 11, 12 and 14.

Large amounts of convection were present in the Southern Hemispheric portion of the Warm Pool region around the time of ATTREX RF05. A series of tropical cyclones are shown in the synoptic analysis chart in figure 9: tropical cyclone Gillian in the Gulf of Carpentaria, tropical cyclone Hadi near the east coast of Queensland, and Tropical Storm Lusi which was intensifying to become a tropical cyclone near the Solomon Islands on 10 March.

[Figure]

**Figure 9.** Synoptic chart from ECMWF ERA-Interim data from 10 March at 00:00 UTC. The three tropical cyclones are labelled, along with their central minimum pressure.

[Figure]

**Figure 11.** Means and 1$\sigma$ standard errors of UCATS ozone along the different sections of RF05. Shading denotes profiles in the Southern Hemisphere.

Before examining the Global Hawk ozone measurements in a meteorological context, account needs to be taken of noise in the UCATS data. Ozone measurements from the level sections (figure 8) were first examined to determine the random measurement error. The combined histogram of the departures from the mean along each level section is shown in figure 10; this approximates well to a Gaussian distribution with standard deviation 3.69 ppbv, suggesting that instrumental random errors in UCATS may be reduced by averaging the data.

[Figure]

**Figure 10.** Histogram of departures of UCATS ozone measurements from the mean along each level section of flight.

As the standard deviation is so large compared with the background ozone values in low-ozone bubbles ($< 15$ ppbv), there is little to be gained from trying to identify individual structures in the ascent and descent profiles. Instead, the approach used here is to average all the data in the troposphere above 14 km along an individual ascent or descent. This requires a definition of the tropopause, taken to be the lowest altitude above which ozone increases by 20 ppbv within a 5 hPa span, and continues to increase thereafter to >50 ppbv. The result of averaging the data along each flight section in this way is shown in figure 11. To calculate the error bars, consideration needs to be given to real variations of ozone along the profiles, which affect the statistical independence of the measurement points. The method described by Wilks

(1995, p.127) was used to calculate the effective number of independent points $N'$:

$$N' = \frac{N(1 - \rho_1)}{(1 + \rho_1)}$$

where N is the number of data points along a section and $\rho_1$ is the lag-1 autocorrelation of the data along that section. The standard error in the mean was then calculated as $\sigma/\sqrt{N'}$, where $\sigma$ was the standard deviation for that section.

The three sets of points shown in figure 11 are independent of one another, and show a consistent pattern. For the first three sets, all the means were above 20 ppbv, with a sharp fall to $< 15$ ppbv in set 4. The means along the level sections remained below 15 ppbv thereafter, but the ascent and descent averages show an increase towards the end of the flight. As expected given the slower ascents and faster descents, error bars for the descents tend to be larger than for the ascents for most of the profiles. Averages along the ascent profiles therefore provide the most precise and representative measures of the mean TTL ozone concentration along the flight path, with a typical standard error of 2 ppbv. We now examine these means in a meteorological context.

[revised manuscript text omitted]

RF06 flew north into the extra-tropics where ozone concentrations are significantly higher, and is therefore not reproduced here.

In summary, an examination of the ATTREX flight data found mean upper tropospheric ozone concentrations as low as 10 ppbv in the outflow of cyclone Lusi in the Southern Hemisphere during flight RF05, but a corresponding flight in the Northern Hemisphere in the outflow of cyclone Faxai found the lowest mean ozone concentration to be 17.5 ppbv. Meanwhile, the FAAM aircraft measured boundary layer

[Figure]

**Figure 17.** Flight track of RF03 with each profile coloured by mean tropospheric ozone as in figure 12. Cyclone Faxai was situated at $\sim(20°N, 150°E)$ during this flight (see figure 5).

[Figure]

**Figure 18.** Flight track of RF04 with each profile coloured by mean tropospheric ozone as in figure 12.

[revised manuscript text omitted]

In the cases of $(CH_3)_2S$, $CH_3I$, $CHBr_3$, $CHBr_2Cl$ and $CH_2Br_2$, all show higher concentrations of the molecule in question when ozone concentrations were less than 20 ppbv

(low-ozone regime) compared to when ozone concentrations were greater than 20 ppbv (high-ozone regime), which suggests that the more ozone-deficient air has encountered the marine boundary layer—where these molecules were produced—more recently than the more ozone-rich air. Meanwhile, $CH_2BrCl$ shows no difference between the low-ozone regime and the high-ozone regime because of its longer lifetime than the other VSLSs measured.

The 33 species that were measured by just the CONTRAST and ATTREX aircraft are plotted in the supplementary material. The species that were of a marine origin show the expected enhancements in the low-ozone regime compared to the high-ozone regime, while the species that were of industrial origin, show the reverse: enhancements were found in the high-ozone regime instead.

[revised manuscript text omitted]

Petropavlovskikh, I., Ray, E., Davis, S. M., Rosenlof, K., Manney, G., Shetter, R., Hall, S. R., Ullmann, K., Pfister, L., Hair, J., Fenn, M., Avery, M., & Thompson, A. M. (2010). Low-ozone bubbles observed in the tropical tropopause layer during the TC4 campaign in 2007. *Journal of Geophysical Research: Atmospheres*, 115(D10).

Quack, B., Peeken, I., Petrick, G., & Nachtigall, K. (2007). Oceanic distribution and sources of bromoform and dibromomethane in the Mauritanian upwelling. *Journal of Geophysical Research: Oceans*, 112(C10).

Rex, M., Wohltmann, I., Ridder, T., Lehmann, R., Rosenlof, K., Wennberg, P., Weisenstein, D., Notholt, J., Krüger, K., Mohr, V., & Tegtmeier, S. (2014). A tropical West Pacific OH minimum and implications for stratospheric composition. *Atmospheric Chemistry and Physics*, 14(9), 4827–4841.

Ridley, B. A., Grahek, F. E., & Walega, J. G. (1992). A Small High-Sensitivity, Medium-Response Ozone Detector Suitable for Measurements from Light Aircraft. *Journal of Atmospheric and Oceanic Technology*, 9(2), 142–148.

Ryall, D. B., Derwent, R. G., Manning, A. J., Simmonds, P. G., & O'Doherty, S. (2001). Estimating source regions of European emissions of trace gases from observations at Mace Head. *Atmospheric Environment*, 35(14), 2507–2523.

Schauffler, S. M., Atlas, E. L., Blake, D. R., Flocke, F., Lueb, R. A., Lee-Taylor, J. M., Stroud, V., & Travnicek, W. (1999). Distributions of brominated organic compounds in the troposphere and

lower stratosphere. *Journal of Geophysical Research: Atmospheres*, 104(D17), 21513–21535.

Singh, H. B., Gregory, G. L., Anderson, B., Browell, E., Sachse, G. W., Davis, D. D., Crawford, J., Bradshaw, J. D., Talbot, R., Blake, D. R., Thornton, D., Newell, R., & Merrill, J. (1996). Low ozone in the marine boundary layer of the tropical Pacific Ocean: Photochemical loss, chlorine atoms, and entrainment. *Journal of Geophysical Research: Atmospheres*, 101(D1), 1907–1917.

Smythe-Wright, D., Boswell, S. M., Breithaupt, P., Davidson, R. D., Dimmer, C. H., & Eiras Diaz, L. B. (2006). Methyl iodide production in the ocean: Implications for climate change. *Global Biogeochemical Cycles*, 20(3).

Stein, A. F., Draxler, R. R., Rolph, G. D., Stunder, B. J. B., Cohen, M. D., & Ngan, F. (2015). NOAA's HYSPLIT Atmospheric Transport and Dispersion Modeling System. *Bulletin of the American Meteorological Society*, 96(12), 2059–2077.

Stemmler, I., Hense, I., & Quack, B. (2015). Marine sources of bromoform in the global open ocean—global patterns and emissions. *Biogeosciences*, 12(6), 1967–1981.

Vömel, H. & Diaz, K. (2010). Ozone sonde cell current measurements and implications for observations of near-zero ozone concentrations in the tropical upper troposphere. *Atmospheric Measurement Techniques*, 3(2), 495–505.

Wilks, D. S. (1995). *Statistical Methods in the Atmospheric Sciences*, chapter 6, (pp. 127). Academic Press, first edition.

Wofsy, S. C. (2011). HIAPER Pole-to-Pole Observations (HIPPO): fine-grained, global-scale measurements of climatically important atmospheric gases and aerosols. *Philosophical Transactions of the Royal Society of London A: Mathematical, Physical and Engineering Sciences*, 369(1943), 2073–2086.

**1 Principal WAS chemicals split by aircraft**

The following three plots show panels of the six chemicals shown in figure 21 of the accompanying article, split into the three individual aircraft. Figure S1 shows the same panel as figure 21 for just the ATTREX data only, figure S2 shows the panel for just the CONTRAST data only, and figure S3 shows the panel for the CAST aircraft data only. In each figure, the average profile for the high-ozone case ($>20$ ppbv) is shown in red and for the low-ozone case ($<20$ ppbv) is shown in blue; the solid lines show the averages for just that aircraft, while the dashed lines show the averages for all the aircraft combined. The amount of CAST aircraft data is small in comparison to CONTRAST and ATTREX and so the effect on the overall averages in figure 21 is negligible. There is some overlap between ATTREX and CONTRAST: the highest altitude that the CONTRAST WAS samples were taken at was $\sim 150$ hPa, and the lowest altitude that the ATTREX WAS samples were taken at was $\sim 180$ hPa.

The ozone measurements taken on board the Gulfstream V aircraft in the CONTRAST campaign were of higher confidence than those taken on board the Global Hawk aircraft in the ATTREX campaign (see section 4.2 of the accompanying article for details on the uncertainties associated with the UCATS ozone measurements from the Global Hawk). However, the differences between the low-ozone cases and the high-ozone cases exist in both the CONTRAST and ATTREX data.

**2 More WAS sample chemicals**

The following plots are of chemical species measured by the whole air samplers (WAS) that were not plotted in the accompanying article. Firstly, dichloromethane ($CH_2Cl_2$) and trichloromethane ($CHCl_3$) were measured by all three aircraft, but unlike the other six chemical species measured by all three aircraft, they both have a strong anthropogenic industrial source with relatively long lifetimes of around five months and six months respectively [Montzka et al., 2010; Carpenter et al., 2014; Khalil and Rasmussen, 1999]. Figure S4 shows the vertical profile of dichloromethane coloured by ozone concentration, with average profiles for WAS samples with ozone concentrations greater than 20 ppbv as a red line, and for WAS samples with ozone concentrations less than 20 ppbv as a blue line, in the same way as the panel plot in figure 21 of the accompanying article. Likewise the profile for trichloromethane is found in figure S5.

The remaining plots show chemical species that were not measured by the FAAM BAe 146 of CAST, but were measured by the CONTRAST and ATTREX aircraft, and categorized by their characteristics. Atmospheric lifetime information comes from González Abad et al. [2011]; Rosado-Reyes and Francisco [2007]; Rudolph [2003]; Pike and Young [2009]; Carpenter et al. [2014]; Prinn et al. [1987]; Wallington et al. [1996]; Olaguer [2002]; Rasmussen and Khalil [1983]; Atkinson et al. [1985]; and Brühl et al. [2012].

**2.1 Aliphatic hydrocarbons**

The aliphatic hydrocarbons measured by the CONTRAST and ATTREX WAS were as follows:

- ethane ($CH_3CH_3$):
  lifetime = $\sim 2$ months (figure S6),

- ethyne ($CH \equiv CH$):
  lifetime = $\sim 2$–4 weeks (figure S7),

- propane ($CH_3CH_2CH_2CH_3$):
  lifetime = $\sim 2$ weeks (figure S8),

- methylpropane ($CH_3CH(CH_3)_2$):
  lifetime = $\sim 1$ week (figure S9),

- butane ($CH_3CH_2CH_2CH_3$):
  lifetime = $\sim 5$ days (figure S10),

- 2-methylbutane ($CH_3CH_2CH(CH_3)_2$):
  lifetime = 4 days (figure S11),

- pentane ($CH_3CH_2CH_2CH_2CH_3$):
  lifetime = $\sim 3$ days (figure S12)

- isoprene ($CH_2 = C(CH_3)CH = CH_2$):
  lifetime = $\sim$minutes–hours (figure S13)

All the hydrocarbons, with the exception of isoprene, follow a similar pattern with enhanced levels of each in the boundary layer when ozone concentrations were high. The difference diminishes with altitude, and at high altitudes, the difference between the low-ozone régime and the high-ozone régime becomes negligible.

Isoprene, however is a naturally occurring chemical emitted in large quantities by vegetation rather than as a result of the petroleum industry, which accounts for the difference between the other hydrocarbons and isoprene.

**2.2 Haloaliphatic compounds**

The haloaliphatic compounds, including chlorofluorocarbons (CFCs), hydrofluorocarbons (HFCs), hydrochlorofluorocarbons (HCFCs) and halons, measured by the CONTRAST and ATTREX WAS were as follows:

**2.2.1 CFCs**

- CFC-12 ($CCl_2F_2$):
  [dichlorodifluoromethane]
  lifetime = $\sim 100$ years (figure S14)

- CFC-11 ($CCl_3F$):
  [trichlorofluoromethane]
  lifetime = $\sim 50$ years (figure S15)

- CFC-112 ($CCl_2FCCl_2F$):
  [tetrachloro-1,2-difluoroethane]
  lifetime = $\sim 60$ years (figure S16)

- CFC-112a ($CCl_3CClF_2$):
  [tetrachloro-1,1-difluoroethane]
  lifetime = $\sim 50$ years (figure S17)

- CFC-113 (CCl$_2$FCClF$_2$)
  [1,1,2-trichloro-1,2,2-trifluoroethane]
  lifetime = $\sim$90 years (figure S18)

- CFC-114 (CClF$_2$CClF$_2$)
  [1,2-dichlorotetrafluoroethane]
  lifetime = $\sim$190 years (figure S19)

**2.2.2 HCFCs**

- HCFC-22 (CHClF$_2$)
  [chlorodifluoromethane]
  lifetime = $\sim$12 years (figure S20)

- HCFC-141b (CH$_3$CCl$_2$F)
  [1,1-dichloro-1-fluoroethane]
  lifetime = $\sim$10 years (figure S21)

- HCFC-142b (CH$_3$CClF$_2$)
  [1-chloro-1,1-difluoroethane]
  lifetime = $\sim$18 years (figure S22)

**2.2.3 HFCs**

- HFC-134a (CH$_2$FCF$_3$)
  [1,1,1,2-tetrafluoroethane]
  lifetime = $\sim$14 years (figure S23)

- HFC-365mfc (CH$_3$CF$_2$CH$_2$CF$_3$)
  [1,1,1,3,3-pentafluorobutane]
  lifetime = $\sim$9 years (figure S24)

**2.2.4 Halons**

- Halon-1211 (CBrClF$_2$)
  [bromochlorodifluoromethane]
  lifetime = $\sim$16 years (figure S25),

- Halon-2402 (CBrF$_2$CBrF$_2$)
  [1,2-dibromotetrafluoroethane]
  lifetime = $\sim$30 years (figure S26)

In all of the cases of CFCs, HCFCs, HFCs and halons, very little variation can be seen, and there is no difference between the low-ozone régime and the high-ozone régime. The background values of the majority of them are non-zero, with little variation from the background values observed. All the CFCs, HCFCs, HFCs and halons are industrial chemicals with often extremely long atmospheric lifetimes. It is likely that these chemicals have reached homogeneity in the atmosphere such that there is little difference between the clean low-ozone régime and the polluted high-ozone régime.

**2.2.5 Others**

- chloromethane (CH$_3$Cl)
  lifetime = $\sim$12 months (figure S27),

- bromomethane (CH$_3$Br)
  lifetime = $\sim$9 months (figure S28),

- 1,1,1-trichloroethane (CH$_3$CCl$_3$)
  lifetime = $\sim$6 years (figure S29),

- tetrachloromethane (CCl$_4$)
  lifetime = $\sim$26 years (figure S30),

- 1,2-dichloroethane (CH$_2$ClCH$_2$Cl)
  lifetime = $\sim$3 months (figure S31),

- trichloroethene (CHCl$=$CCl$_2$)
  lifetime = $\sim$5 days (figure S32),

- tetrachloroethene (CCl$_2$$=$CCl$_2$)
  lifetime = $\sim$5 months (figure S33)

All of these chemicals are produced industrially. Chloromethane, bromomethane and 1,2-dichloroethane have the expected profiles for anthropogenic chemicals—the polluted, high-ozone régime is enhanced compared to the clean, low-ozone régime. However, 1,1,1-trichloroethane and tetrachloromethane are the opposite way round; their lifetimes are particularly long, similar to the lifetimes of the CFC, HFC, HCFC and halon groups. Both trichloromethane and tetrachloroethene show large enhancements in the high-ozone régime in the boundary layer, but in the mid-troposphere there is an unexpected enhancement of each in the low-ozone régime.

**2.3 Aromatic compounds**

- benzene (C$_6$H$_6$)
  lifetime = $\sim$months (figure S34)

- chlorobenzene (C$_6$H$_5$Cl)
  lifetime = $\sim$2 weeks (figure S35)

Benzene and chlorobenzene are industrial solvents, and both show enhancements in the high ozone régime compared to the low ozone régime, which is what is expected. However, in the mid-troposphere, chlorobenzene shows the opposite.

**2.4 Sulfides**

- carbonyl sulfide (OCS)
  lifetime = $\sim$35 years (figure S36)

Like dimethyl sulfide, shown in figure 15 of the accompanying article, carbonyl sulfide, shows a slight enhancement in the low-ozone, clean régime.

[Figure]

Figure S1: Panel of the six principal WAS chemicals using the ATTREX WAS sample data only.

[Figure]

Figure S2: Panel of the six principal WAS chemicals using the CONTRAST WAS sample data only.

[Figure]

Figure S3: Panel of the six principal WAS chemicals using the CAST aircraft WAS sample data only.

[Figure]

Figure S4: Dichloromethane

[Figure]

Figure S7: Ethyne

[Figure]

Figure S5: Trichloromethane

[Figure]

Figure S8: Propane

[Figure]

Figure S6: Ethane

[Figure]

Figure S9: Methylpropane

[Figure]

Figure S10: Butane

[Figure]

Figure S13: Isoprene

[Figure]

Figure S11: 2-Methylbutane

[Figure]

Figure S14: CFC-12

[Figure]

Figure S12: Pentane

[Figure]

Figure S15: CFC-11

[Figure]

Figure S16: CFC-112

[Figure]

Figure S19: CFC-114

[Figure]

Figure S17: CFC-112a

[Figure]

Figure S20: HCFC-22

[Figure]

Figure S18: CFC-113

[Figure]

Figure S21: HCFC-141b

[Figure]

Figure S22: HCFC-142b

[Figure]

Figure S25: Halon 1211

[Figure]

Figure S23: HFC-134a

[Figure]

Figure S26: Halon 2402

[Figure]

Figure S24: HFC-365mfc

[Figure]

Figure S27: Chloromethane

[Figure]

Figure S28: Bromomethane

[Figure]

Figure S29: 1,1,1-trichloroethane

[Figure]

Figure S30: Tetrachloromethane

[Figure]

Figure S31: 1,2-dichloroethane

[Figure]

Figure S32: Trichloroethene

[Figure]

Figure S33: Tetrachloroethene

[Figure]

Figure S34: Benzene

[Figure]

Figure S35: Chlorobenzene

[Figure]

Figure S36: Carbonyl sulfide

> We present six of these molecules here—dichloromethane and trichloromethane, as well as the species measured by the CONTRAST and ATTREX aircraft but not the CAST aircraft are plotted in the supplementary material. Dichloromethane is a predominantly industrially produced chemical with a strong anthropogenic signal and relatively long lifetime ($\sim$6 months), while trichloromethane has mostly natural, but some anthropogenic, sources [?], and has a long lifetime of $\sim$6 months.

and paragraphs 6 and 7 of section 6 now read:

> In the cases of $(CH_3)_2S$, $CH_3I$, $CHBr_3$, $CHBr_2Cl$ and $CH_2Br_2$, all show higher concentrations of the molecule in question when ozone concentrations were less than 20 ppbv (low-ozone regime) compared to when ozone concentrations were greater than 20 ppbv (high-ozone regime), which suggests that the more ozone-deficient air has encountered the marine boundary layer—where these molecules were produced—more recently than the more ozone-rich air. Meanwhile, $CH_2BrCl$ shows no difference between the low-ozone regime and the high-ozone regime because of its longer lifetime than the other VSLSs measured.
>
> The 33 species that were measured by just the CONTRAST and ATTREX aircraft are plotted in the supplementary material. The species that were of a marine origin show the expected enhancements in the low-ozone regime compared to the high-ozone regime, while the species that were of industrial origin, show the reverse: enhancements were found in the high-ozone regime instead.

2. A coding issue was discovered, which erroneously removed much of the ozone data from the FAAM aircraft. The affected figures were figure 19, the bottom plot of figure 20, and figure 21, and have now been plotted with the correct data-set.

3. p. 3, l. 40 "The [ATTREX] instruments were calibrated on the ground against a NIST-certified calibration system before and after the mission" rather than before and after every flight.

---

## Referee Report (RR1)

Summary review of *Newton et al. (revised 2018)*
"Observations of ozone-poor air in the Tropical Tropopause Layer"

I recommended that the first version of this paper be accepted subject to major revisions.  I had four major concerns:

- There needed to be a more thorough analysis of the UCATS ozone data and uncertainties in the average values presented in the paper
- The possible impact of inter-aircraft ozone biases and their effects on the presentation of the WAS data in (the original) Fig. 17
- The paper needed a more detailed background section, and
- It needed a clearer presentation of a connection between the science questions raised in the introduction to the datasets brought into the analysis.

The revised manuscript more than successfully addresses all of these concerns, and I am very happy to recommend it for publication.   I have rated it Excellent in each of the three categories of scientific significance, scientific quality, and presentation quality.